# Practice Patterns of Antithrombotic Therapy during the Early Postoperative Course of Cardiac Surgery

**DOI:** 10.3390/jcm12052029

**Published:** 2023-03-03

**Authors:** Thomas Klein, Hugo Bignolas, Nicolas Mongardon, Osama Abou-Arab, Pierre Grégoire Guinot, Adrien Bouglé, Philippe Guerci

**Affiliations:** 1Department of Anesthesiology and Critical Care Medicine, Cardiothoracic and Vascular Anesthesia, University Hospital of Nancy-Brabois, 54511 Vandoeuvre-lès-Nancy, France; 2INSERM 1116, DCAC, University of Lorraine, 54500 Vandoeuvre-lès-Nancy, France; 3Service d’Anesthésie-Réanimation Chirurgicale, DMU CARE, DHU A-TVB, Assistance Publique-Hôpitaux de Paris (AP-HP), Hôpitaux Universitaires Henri Mondor, 94010 Créteil, France; 4U955-IMRB, Equipe 03 “Pharmacologie et Technologies pour les Maladies Cardiovasculaires (PROTECT)”, Inserm, Univ Paris Est Créteil (UPEC), Ecole Nationale Vétérinaire d’Alfort (EnVA), 94700 Maisons-Alfort, France; 5Faculté de Santé, Université Paris Est Créteil, 94010 Créteil, France; 6Department of Anaesthesiology and Critical Care Medicine, Amiens University Medical Centre, 80000 Amiens, France; 7Department of Anaesthesiology and Critical Care Medicine, Dijon University Medical Centre, 21000 Dijon, France; 8LNC UMR1231, University of Burgundy and Franche-Comté, 21000 Dijon, France; 9GRC 29, Assistance Publique-Hôpitaux de Paris (AP-HP), Sorbonne Université, 75013 Paris, France; 10DMU DREAM, Département d’anesthésie et réanimation, Institut de Cardiologie, Hôpital La Pitié-Salpêtrière, 75013 Paris, France

**Keywords:** anticoagulation, low-molecular-weight heparin, cardiac surgery, mechanical heart valve replacement, thromboprophylaxis

## Abstract

Background: The current practices regarding the management of antithrombotic therapy during the early postoperative course of cardiac surgery are not well described. Methods: An online survey with multiple-choice questions was sent to cardiac anesthesiologists and intensivists from France. Results: The response rate was 27% (n = 149), with 2/3 of the respondents having less than 10 years of experience. A total of 83% of the respondents reported using an institutional protocol for antithrombotic management. A total of 85% (n = 123) of the respondents regularly used low-molecular-weight heparin (LMWH) during the immediate postoperative course. For 23%, 38%, 9%, and 22% of the physicians, LMWH administration was initiated between the 4th and 6th hour, between the 6th and 12th hour, between the 12th and 24th hour, and on postoperative day 1, respectively. The main reasons for not using LMWH (n = 23) were a perceived increased risk of perioperative bleeding (22%), poor reversal compared with unfractionated heparin (74%), local habits and the refusal of surgeons (57%), and its overly complex management (35%). The modalities of LMWH use were widely varied among the physicians. Chest drains were mostly removed within 3 days of surgery with an unchanged dose of antithrombotic therapy. Regarding temporary epicardial pacing wire removal anticoagulation, 54%, 30%, and 17% of the respondents left the dose unchanged, suspended the anticoagulation, or lowered the anticoagulation dose, respectively. Conclusion: LMWH was inconsistently used after cardiac surgery. Further research is warranted to provide high-quality evidence regarding the benefits and safety of LMWH use early after cardiac surgery.

## 1. Introduction

Cardiac surgery carries one of the narrowest therapeutic windows regarding the use of antithrombotic therapy during the postoperative course. Postcardiac surgery patients are exposed to an increased risk of thrombotic events secondary to inflammation after cardiopulmonary bypass and/or baseline hypercoagulability and hemorrhagic events due to coagulation disorders, extensive surgery, hyperfibrinolysis, or antithrombotic therapies [1,2,3,4]. Although most guidelines recommend bridging oral anticoagulation (OAC) with low-molecular-weight heparin (LMWH) during the preoperative period [5], the data are inconsistent regarding the type of antithrombotic, the dosage, and the starting time during the early postoperative period, and there is scarce evidence supporting the use of specific strategies. Although some observational studies on the use of LMWH are reassuring, the systematic reviews and meta-analyses of the use of LMWH after mechanical heart valve replacement remain inconclusive [6,7,8,9,10,11,12,13,14]. Regarding coronary artery bypass grafting (CABG), the rates of venous thromboembolism (VTE) and bleeding are not different between LMWH (enoxaparin) and unfractionated heparin (UFH) [15,16,17,18]. More than 20 years after the first publications describing the use of LMWH during the early postoperative course of cardiac surgery, the debate is still raging. The use of LMWH in the context of early recovery after cardiac surgery may be of interest, as it requires less biological monitoring and has a simplified route of administration.

The degree to which emerging evidence from the literature has been translated to clinical cardiac anesthesia practice is unknown. We hypothesized that the practice patterns may widely vary among centers and habits, and that they may not meet the current safety guidelines for the use of LMWH.

Thus, we sought to investigate the real-life use of LMWH in the context of early postoperative cardiac surgery among French cardiac anesthesiologists and intensivists through an online survey. This survey was aimed at investigating four areas of interest: (1) the frequency of LMWH use and the type of surgery; (2) the timing and dosage of the antithrombotic administration (including bridging with OAC); (3) the go/no-go for antithrombotic administration; (4) the management of chest drain and temporary epicardial pacing wire (TEPW) removal with regard to antithrombotic therapy.

## 2. Materials and Methods

### 2.1. Population

The survey population was defined as physician members who were currently practicing cardiac anesthesia from the ARCOTHOVA group (French Anesthesiologists, Intensivists and Surgeons with practice in Cardiothoracic and Vascular Surgery) and SFAR (French Society of Anesthesia and Intensive Care Medicine) research network. Residents were excluded. No consent was required to participate in the online survey. Each network was asked to send an email invitation with the link to the online survey to their qualifying members, which was followed by 2 reminders.

### 2.2. Online Survey and Periods

The full survey took approximately ten minutes to complete, and it comprised a maximum of 40 questions, most of which were multiple-choice questions. The survey questions were designed by two authors (TK and PG), and they were distributed for comments and modifications within the study group. Practices related to the management of transcatheter aortic/mitral valve replacement (TAVR/TMVR) and structural cardiology were excluded. The survey was posted online (Google Forms survey). We did not track participant-identifying information to ensure anonymity. The participants were not given the option of skipping questions, and each questionnaire had to be entirely completed before being submitted or it was not considered for analysis. Including the different waves of invitations and reminders to answer the survey, the data were collected from April to July 2022.

### 2.3. Data Analysis

Most of the variables were categorical and were expressed as frequencies and percentages. Multiple answers were allowed for several questions, so the cumulative percentages presented in the text, or figures, exceed 100%. The analysis was performed using Prism 9.5 GraphPad Software, La Jolla, CA, USA.

The sample size was calculated based on the estimated membership sizes of the cardiac anesthesiologists/surgeons of the ARCOTHOVA group and/or SFAR. For the sensitivity power analysis, the authors assumed a total number of ~550 physicians and a response rate of 25–30%. A sample size from 145 to 225 would produce a margin of error of 5–7% for 95% confidence intervals.

## 3. Results

### 3.1. Respondent Characteristics

A total of 149 physicians (149/550 = 27%) responded to the questionnaire. The characteristics of the respondents are provided in Table 1. A total of 144 respondents were anesthesiologists/intensivists (97%), and the remainder were cardiac surgeons. The responses came from all over France, with a homogeneous repartition, and the Auvergne-Rhône-Alpes region was the most represented (n = 27, 18%). A total of 20% of the respondents practiced in private facilities. Two-thirds of the respondents had less than 10 years of experience in cardiac surgery.

### 3.2. Early Postoperative Antithrombotic Therapy

Most of the respondents (83%) reported using an institutional antithrombotic management protocol after on-pump CABG or valve replacement (TAVR excluded) and primarily using an anticoagulant in addition to antiplatelet therapy (not antiplatelets alone). In patients with normal postoperative coagulation tests and/or no excessive bleeding from chest tubes, the administration of the anticoagulant (LMWH or UFH) was started between the 4th and 6th hours, between the 6th and 12th hours, between the 12th and 24th hours, or on postoperative day (POD) + 1 for 49%, 44.3%, 6%, and 6.7% of the respondents, respectively. Regarding the postoperative coagulation tests, most of the respondents (61.5%) were likely to begin anticoagulation when the prothrombin ratio, which is defined as the plasma prothrombin time divided by the mean normal prothrombin time, was between 50 and 65%. Interestingly, 23% of the respondents began anticoagulants with a prothrombin ratio between 30 and 50%. Similarly, ratios of activated partial thromboplastin time (r-aPTT) of <1.5 and <2.0 were considered desirable for beginning anticoagulants for 37.2% and 36.6% of the respondents, respectively. A total of 50% of the respondents considered a platelet count of >50 G/L to be sufficient, while 34.2% preferred to administer anticoagulants when the threshold was >75 G/L. Interestingly, from 15 to 20% of the physicians assessed chest tube drainage and hemodynamic stability before beginning anticoagulation, regardless of the actual values of the biological parameters.

### 3.3. Modalities of LMWH Use

A total of 85% (n = 126) of the respondents regularly used LMWH during the immediate postoperative course. For 23%, 38%, 9%, and 22% of the physicians, LMWH administration was initiated between the 4th and 6th hour, between the 6th and 12th hour, between the 12th and 24th hour, and on POD + 1, respectively. In cases of prosthetic valvular surgery, irrespective of the type of valve (mechanical or bioprosthetic), LMWH was administered at the standard thromboprophylaxis dose for 83% of the respondents, at a therapeutic dose for 10% of the respondents, and at an intermediate dose for 7% of the respondents. Interestingly, approximately 40% of the physicians first started with UFH for a few days in the cardiac intensive care unit (ICU) before switching to LMWH. When a therapeutic dose of UFH was deemed necessary, more than half of the physicians started at the thromboprophylaxis dose, with a step increase over 1 or 2 days to reach the desired dose.

For most of the respondents, the use of LMWH remained contraindicated in patients with creatine clearances < 30 mL·min^−1^·m^−2^. However, 36% of the physicians did not use LMWH when the creatine clearance was below 50 mL·min^−1^. Enoxaparin was the LMWH almost exclusively used by the physicians.

Figure 1 summarizes the different types of cardiac surgeries for which LMWH was prescribed. Combined surgeries are not individualized. For postoperative mechanical aortic or mitral valve replacement, the use of UFH was preferred over LMWH. Of the 126 respondents who declared that they used LMWH during postoperative cardiac surgery, 70 (56%) did not use it for mechanical aortic valve replacements, and 78 (62%) did not use it for mechanical mitral valve replacements. In cases of postoperative atrial fibrillation while on LMWH at a thromboprophylaxis dose, 62% of the respondents increased the dose, and 27% of them switched it for a therapeutic dose with UFH, either intravenously or subcutaneously.

The main reasons given by the physicians for not using LMWH (n = 23) were a perceived increased risk of perioperative bleeding (22%), poor reversal compared with UFH (74%), local habits and/or the refusal of surgeons (57%), and its overly complex management (35%). Interestingly, even though the use of LMWH is not regulatory-agency-approved in France, in this setting (i.e., mechanical prosthesis), it was considered a hindrance for only a few respondents (n = 3). Among the respondents, 68% started vitamin K antagonists (VKAs) in the early postoperative setting when medically indicated, with the majority (28%) starting on POD + 5 (Figure 2). For many of the physicians, this was largely dependent on the absence of bleeding and on the chest tube and TEPW removal, which explains why the majority (47% of respondents) of them stated that they begin oral anticoagulation in the surgical ward, after most patients have already been discharged from the ICU on POD 3.

### 3.4. Management of Chest Drainage and Temporary Epicardial Pacing Wires (TEPWs)

Figure 3 depicts the current timing of the chest drainage removal according to the type of surgery. Most of the respondents removed the drains early (<3 days), even in aortic root surgeries. At the time of the chest pericardial and pleural drain removal, the dose was left unchanged, regardless of the dosage, for 72% and 80% of the respondents, respectively.

Almost 90% of the physicians proceeded to TEPW removal. In this case, the practice patterns were inconsistent: 53.8% (n = 78) left the dose unchanged, 30% (n = 43) suspended the anticoagulation before the removal, and 17% (n = 24) lowered the anticoagulation dose regardless of the type (UFH or LMWH).

## 4. Discussion

The purpose of this nationwide survey was to assess the current management of early postoperative anticoagulation after on-pump cardiac surgery by French cardiac anesthesiologists and intensivists. This survey highlights a diverging practice, with considerable heterogeneity in the type and timing of the anticoagulation, a lack of homogeneous guidelines, and difficulties in implementing changes regarding anticoagulation strategies in this setting. The management of anticoagulation was inherent to local habits, or to physicians/surgeons who were reluctant to embrace changes to their practices despite the reassuring and consistent literature on the topic.

In the present study, among the 149 respondents, 83% had postoperative anticoagulation protocols for cardiac surgery and initiated anticoagulants early in the postoperative period (mostly within the first 12 postoperative hours). Most of the respondents used LMWH post cardiac surgery. For 60% of the respondents, the use of LMWH was mostly initiated at a prophylactic dose and started immediately. However, when a bridge was deemed necessary, VKA initiation was mostly postponed until the patient left the cardiac ICU.

In 2000, Montalescot et al. reported the first large and comparative series of patients who were anticoagulated with LMWH (n = 106, 100 UI·kg^−1^ of enoxaparin subcutaneously administered twice daily) after mechanical heart valve replacement [11]. The authors demonstrated that LMWH administration during the high-risk postoperative period was at least as safe and effective as UFH administration in a comparable group of patients. Similar observations were made in two other studies, with the associated shorter bridging times to OAC and hospital lengths of stay [9,12]. However, in all the studies, therapeutic LMWH administration was not started in the immediate postoperative period but rather on POD + 5-6.

Subsequently, several studies examined the use of LMWH during the immediate postoperative course, from the 6th hour to POD + 1 after cardiac surgery (Table 2) [7,13,14,15,19,20,21,22,23]. No excessive thromboembolic events were observed; however, there was a trend toward more minor bleeding after surgery.

Kulik et al. reviewed the different anticoagulation strategies after heart surgery, which include the following: (1) oral VKA alone starting on POD + 1; (2) intravenous UFH during the immediate postoperative course and oral VKA on POD + 1; (3) LMWH started early after surgery and oral VKA on day 1 [24]. The authors did not find an excess risk of bleeding or thromboembolic events with the use of LMWH compared with UFH; however, these studies were largely underpowered. These results have been challenged by a larger study that included 2977 patients who underwent cardiac surgery, which showed an increased risk for surgical re-exploration due to bleeding in patients who received UFH or LMWH compared with patients who did not receive either (adjusted HR = 2.5; CI [1.8–3.8]; *p* < 0.001, and adjusted HR = 2.7; CI [1.7–4.2]; *p* < 0.001, respectively) [25]. Of note, almost half the patients received preoperative anticoagulation. In addition, thromboembolic events were not documented. In our survey, most of the respondents preferred to avoid the use of LMWH and favored the administration of UFH for mechanical heart valves. The introduction of oral anticoagulants, when indicated (e.g., mechanical valve prostheses, rhythm disorders), also varied among the respondents. This treatment is introduced at different times and even according to certain circumstances (after the removal of epicardial electrodes, for example). However, in the literature, we did not identify any excess risk of bleeding in cases of the early introduction of VKAs.

Given the complex balance between bleeding and thromboembolic events, postoperative anticoagulation bridging therapy after cardiac surgery remains challenging, especially with regard to the chest drain and TEPW removal. In a retrospective study that included 11,754 patients and that examined the incidence of tamponade following TEPW removal, 8 out of 11 patients were on LMWH (dalteparin at 5000 UI SC daily) plus antiplatelet therapy [26]. Two patients had an unfavorable outcome. This highlights the need for a personalized approach to anticoagulation management. One-size-fits-all approaches may not be suitable. Point-of-care anticoagulation guidance could be a valuable tool for initiating, adjusting, or discontinuing anticoagulation therapy.

According to the product label, enoxaparin, as well as other LMWHs, is not recommended for patients with mechanical prosthetic valves. However, the EACTS/ESC guidelines recommend that “treatment with VKA should be started on the first postoperative day in combination with bridging therapy [with therapeutic doses of either UFH or off-label use of LMWH] until therapeutic INR is achieved” (Class I, Level B) and should be started 12–24 h after surgery (Class I, Level C) [27], which contrasts with the real-life practices in France that are reported in the present survey. A protocol for early LMWH use considering the published evidence is proposed in Appendix A. Direct oral anticoagulants (DOACs) are increasingly used worldwide, and cardiac surgery makes no exception [28]. Several studies have already reported the use of DOACs with bioprosthetic valves instead of VKAs; however, the outcomes are inconsistent [28,29,30]. Regarding TAVR, the use of a DOAC (rivaroxaban) was associated with a higher risk of death or thromboembolic complications, as well as with a higher risk of bleeding, than an antiplatelet-based strategy [31]. The use of DOACs in this setting will increase in the coming years and add further confusion. To date, only one randomized controlled trial (RE-ALIGN) has investigated early DOAC (dabigatran) use for patients with mechanical heart valves, and it was prematurely halted because of increased thromboembolic and bleeding events compared with the use of VKA [32].

Our study has limitations. First, survey results, by nature, are driven by the respondents and may capture only a part of real life. One of the major limitations is that the 27% response rate is only from France, which greatly limits the generalizability of the results. Second, while the survey design attempted to capture all current practice patterns of postoperative anticoagulation after cardiac surgery, all situations may not have been covered. Thus, some questions were left with open answers, which are less easily reportable. Ultimately, it can be challenging to factor in a patient’s individual circumstances, highlighting the necessity for tailored or personalized therapy.

**Table 2 jcm-12-02029-t002:** Characteristics of studies with low-molecular-weight heparin use in the early postoperative course of cardiac surgery.

Author (Year)	Study Type	No. of Patients	Type of Cardiac Surgery	Strategy	Time of Start	Follow-Up	TE Rate	TE Event	Bleeding Rate	Mortality
Montalescot [11] (2000)	Retrospective CC	208 adults 106 adults UFH 102 adults LMWH	Single or double mechanical heart valve replacement	UFH group: 3 SC injections a day, at a dose of 500 IU·kg^−1^·d^−1^, adjusted to the APTT with a target range of 1.5 to 2.5 times control; LMWH group: 100 anti-Xa IU·kg^−1^, SC at 12 h intervals for enoxaparin. For Nadroparin, at a dose of 87 anti-Xa IU·kg^−1^ SC at 12 h intervals. Anti-Xa activity 4 h after the third injection: 0.5 to 1 IU·mL^−1^	~POD + 6	In-hospital stay (~14 days)	UH: 1/106 (1%) LMWH: 0/102 (0%)	2 successive transient ischemic strokes (J17)	UH: 2/106 (2%) LMWH: 2/102 (2%)	No death
Fanikos [9] (2004)	Retrospective matched CC	63 adults 34 adults UFH 29 adults LMWH	Mechanical heart valve	UFH dose adjustments left to the discretion of the physician; Enoxaparin 1 mg/kg SC at 12 h interval	POD + 1-2	3 months	UFH: 2/34 (6%) LMWH: 0/29 (0%)	1 bilateral occipital stroke. 1 HIT	UH: 3/34 (9%) LMWH: 3/29 (10%)	UH: 4/34 (11.7%) LMWH: 1/29 (3.4%)
Talwar [19] (2004)	Retrospective CC	538 adults 245 OA 293 OA + LMWH	Mechanical heart valve	OA + enoxaparin SC	6 h after surgery	6 months	OA: 15 (6.1%) OA + LMWH: 6 (2.1%)	Prosthetic valve thrombosis	N.D.	N.D.
Meurin [12] (2005)	Prospective CS	250 patients	Mechanical heart valve	Enoxaparin (100 IU·kg^−1^ twice daily SC) until VKA treatment was fully effective	~POD + 16	3 months	1/250 (0.4%)	Transient ischemic attack	2/250 major bleeding (0.8%), 3/250 minor bleeding (1.2%)	No death
Jones [25] (2005)	Retrospective CC	2977 adults 2037 none 579 UFH 361 LMWH	CABG or valve surgeries	Enoxaparin: 1 mg·kg^−1^ SC twice daily OA not documented	N.D.	In-hospital stay	N.D.	N.D.	Bleeding requiring reoperation: 2.7% for no UFH or LMWH, 7.8% for UFH 8.9% for LMWH	N.D.
Rivas-Gándara [20] (2008)	Prospective CS	140 adults	Mixed heart valve replacement	Enoxaparin 40 mg SC twice daily if BW > 60 kg or 40 mg once daily if BW < 60 kg, then increase to 0.75 mg·kg^−1^·d^−1^ at POD + 3 OA started at POD + 1	POD + 1	3 months	6/140 (4.3%)	3 strokes, 1 mesenteric ischemia, 1 retinal thrombosis, 1 mitral prosthetic thrombosis	3/140 (2.1%), 1 mortal intracranial hemorrhage, 1 muscular hematoma, 1 excessive drainage bleeding	9/140 (6.4%)
Puri [21] (2008)	CS	503 adults 221 OA only 159 OA + LMWH 123 OA + UFH	Mechanical heart valve	LMWH: Enoxaparin 40 mg SC once daily UFH infusion titrated in order to achieve 1.5- to 2-fold that of controls	6–12 h after surgery	8 to 64 months	OA only: 1/221 (0.5%); OA + LMWH: 1/159 (0.6%); OA + UFH: 1/123 (0.8%)	Transient ischemic attack, peripheral thromboembolism	2/221 (OA), 12/159 (OA + LMWH), 9/123 (OA + UFH) required the reinsertion of drains; 0/221 (OA), 7/159 (OA + LMWH), 5/123 (OA + UFH) had tamponade; 0/221 (OA), 5/159 (OA + LMWH), 4/123 (OA + UFH) required re-exploration for excessive drainage at >48 h after surgery	N.D.
Steger [33] (2008)	Retrospective CS	256 adults	Mechanical heart valve	Enoxaparin 40 mg SC twice daily	POD + 4	Mean follow-up: 38.6 days	2/256 (0.7%)	Arterial thromboses	18/256 (7%) minor bleeding	No death
Weiss [14] (2013)	Retrospective CC	402 adults Enoxaparin 201 full dose (FD) 201 half dose (HD)	Mixed cardiac surgeries	FD: Enoxaparin 1 mg·kg^−1^ HD: Enoxaparin 0.5 mg·kg^−1^	POD + 1	In-hospital stay	FD: 5/201 (2.5%) HD: 9/201 (4.5%)	Cerebral infarction: 1/201 (FD), 5/201 (HD); Myocardial infarction: 3/201 in each group; Mesenteric infarction: 0/201 (FD), 1/201 (HD); Vein thrombosis: 1/201 (FD), O/201 (HD)	FD: 11/201 (5.5%) HD: 5/201 (2.5%)	FD: 1/201 (0.5%) HD: 11/201 (5.5%)
Kindo [10] (2014)	Prospective CS	1063 adults Enoxaparin	Mechanical heart valve	Enoxaparin 4000 IU SC on POD + 1 At POD + 2: 4000 IU SC—BW < 60 kg 6000 IU –SC BW [60–80 kg] 8000 IU SC—BW > 80 kg Twice daily	POD + 1	6 weeks	11/1063 (1%)	10/11 transient or permanent strokes	44/1063 (4.1%) 7 were observed before enoxaparin	No death
Kolluri [15] (2016)	RCT	78 adults 41 fondaprinux 37 placebo	On-pump CABG	2.5 mg SC fondaparinux daily prophylaxis	12 ± 2 h after surgery or morning of POD + 1	POD + 11 And POD + 35	Fondaparinux: 1/41 (2.4%) Placebo: 1/37 (2.7%)	DVT	Fondaparinux: 4/41 (9.8%) Placebo: 0/37	No death
Czerwińska-Jelonkiewicz [13] (2018)	Retrospective CS	388 adults Enoxaparin	Surgical valve procedure Mechanical 161 (62%) Bioprosthetic 110 (28.35%) Other/combined 117 (30.1%)	40–60 mg SC enoxaparin on day of surgery and 40–80 mg on POD + 1	8–12 h after surgery	In-hospital stay	7/388 (1.8%)	Strokes or transient ischemic attacks	Severe: 37 (9.6%) Massive: 14 (3.6%)	4/388 (1.0%)
Li [7] (2019)	Retrospective CS with propensity score	473 adults 257 enoxaparin 216 controls	On-pump minimally invasive cardiac surgery (MICS)	40 mg SC enoxaparin once daily	6 h after surgery	In-hospital stay	Enoxaparin: 1/257 (0.4%); Control: 3/216 (1.4%); After PS adjusting Enoxaparin and Control: 0%	DVT and shock (undetermined)	Enoxaparin: 41/257 (16%); Control: 17/216 (7.9%); After PS enoxaparin: 16/257 (14.5%); Control: 6/216 (5.5%)	No death
Parviainen [23] (2022)	RCT	39 adults Enoxaparin 20 CIV 19 SC	On-pump CABG	40 mg enoxaparin CIV/24 h for 72 h 40 mg SC enoxaparin once daily	6–10 h after surgery	72 h postoperative	0/33	None in both groups	3/39 (7.7%)	No death

BW: body weight; CABG: coronary artery bypass graft surgery; CIV: continuous intravenous infusion; CS: case series; CC: case–control; DVT: deep venous thrombosis; HIT: heparin-induced thrombocytopenia; IU: international unit; LMWH: low-molecular-weight heparin; N.D: not documented; OA: oral anticoagulation; POD: postoperative day; PS: propensity score; RCT: randomized controlled trial; SC: subcutaneous; TE: thromboembolic; UFH: unfractionated heparin.

## 5. Conclusions

The real-life use of LMWH early after cardiac surgery is inconsistent among the practices of French physicians, mainly due to its off-label use, the reluctance of physicians to change, and the fear of increased complications (bleeding and thromboembolic events). High-quality evidence is urgently needed to confirm the safety and benefits of the use of LMWH during the immediate postoperative course of cardiac surgery.

## Figures and Tables

**Figure 1 jcm-12-02029-f001:**
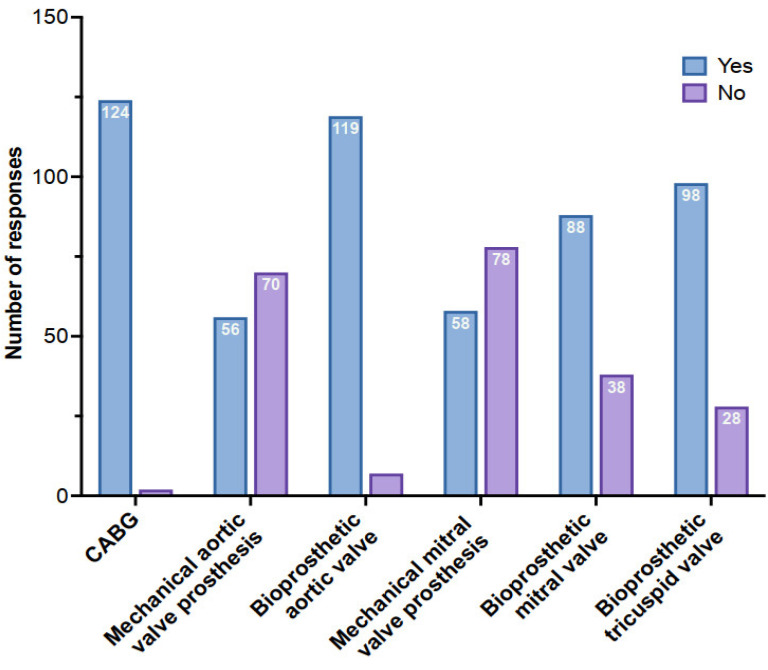
Number of respondents using low-molecular-weight heparin according to the type of cardiac surgery. CABG: coronary artery bypass graft.

**Figure 2 jcm-12-02029-f002:**
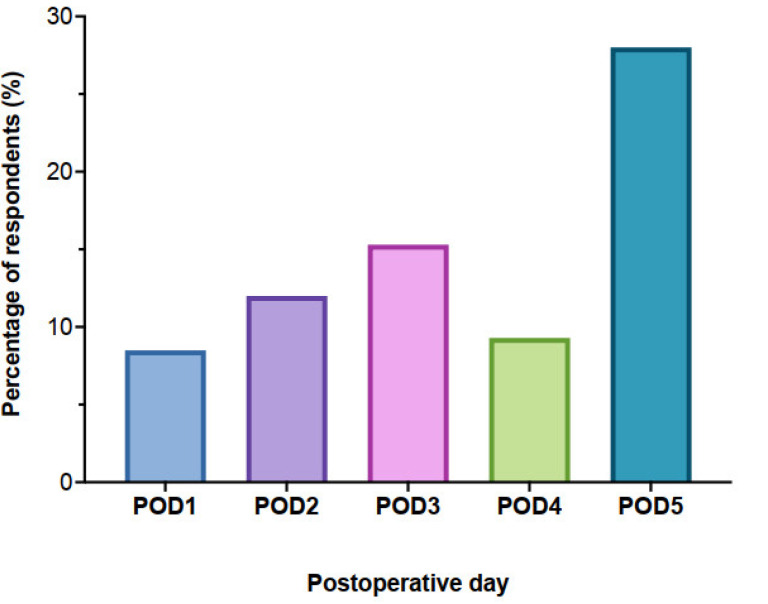
Timing of vitamin K antagonist initiation after cardiac surgery. POD: postoperative day.

**Figure 3 jcm-12-02029-f003:**
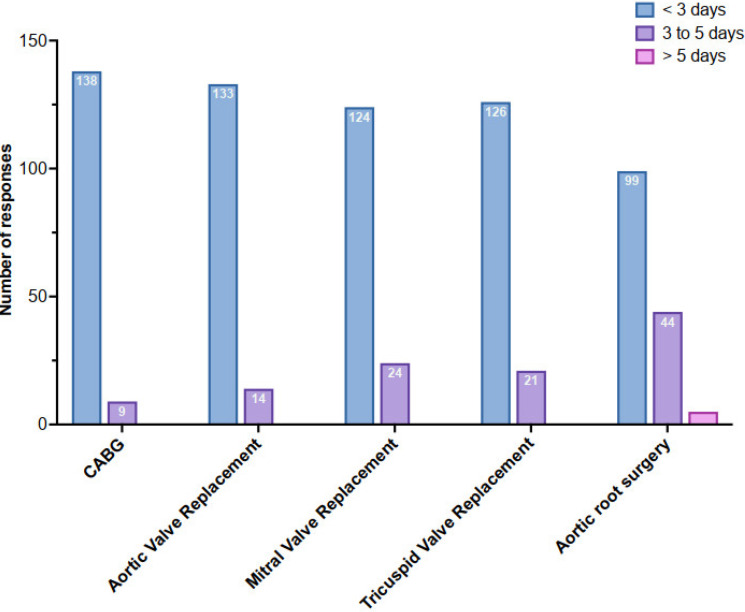
Collected responses regarding the timing of chest/pericardial drain removal according to cardiac surgery procedure.

**Table 1 jcm-12-02029-t001:** Characteristics of the respondents.

Characteristics	Number of Responses (n (%))
Total population	149
Age (years)	
<30	3 (2)
30 to 40	88 (59.1)
40 to 50	31 (20.8)
>50	27 (18.1)
Years of practice in cardiac anesthesia	
<5	70 (47)
6 to 10	29 (19.5)
11 to 15	27 (18.1)
16 to 20	8 (5.4)
>20	15 (10.1)
Position in the healthcare facility	
Professor or associate professor	10 (6.7)
Attending physician	114 (76.5)
Resident	25 (16.8)
Type of hospital	
Academic hospital	116 (77.9)
Nonacademic public hospital	5 (3.4)
For-profit private facility	21 (14.1)
Non-profit private facility	7 (4.6)
Minimally invasive cardiac surgery (MICS) in the center	
Yes	106 (71.1)
No	43 (28.9)
Minimal extracorporeal circulation (MECC)	
Yes	20 (13.5)
No	128 (86.5)
Volume of on-pump heart surgery (cases per year)	
250 to 499	26 (17.4)
500 to 749	42 (28.2)
750 to 1000	34 (22.8)
>1000	47 (31.5)

Data are expressed as frequencies and percentages.

## Data Availability

On reasonable request.

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
