# Peer review of "Practice Patterns of Antithrombotic Therapy during the Early Postoperative Course of Cardiac Surgery"

_jcm, 2023, doi:10.3390/jcm12052029_

Round 1
Reviewer 1 Report
Very good paper whose purpose was to assess the current management of early postoperative anticoagulation after on-pump cardiac surgery by French Cardiac Anesthesiologists and Intensivists. This survey highlights a diverging practice, with considerable heterogeneity in the type and timing of anticoagulation, the lack of homogeneous guidelines and the difficulties in implementing changes regarding anticoagulation strategies in this setting.
Very good methods used, and heterogeneity of data. The study is easy to read and very comprehensive.
The authors conclude that Real life use of LMWH early after cardiac surgery is inconsistent among the French anesthesiologists mainly due its off-label use, reluctancy to change and fear of increased complications .
Author Response
We would thank Reviewer #1 for his/her kind comments. We hope that this study will promote research in this field.
Reviewer 2 Report
The authors describe an online survey sent to anaesthesiologists and intensivists in France, over a 3 month period, regarding their opinion on a number of questions related to the use of LMWH in patients in the post operative period following cardiac surgery. The authors wished to obtain such opinions as the evidence regarding how anticoagulation should be managed in this period is unclear
The methodology is appropriate and well described and the study does indicate divergent practice among the small sample size that was assessed as a proportion of the number of professionals sent questionnaires (27% of the cohort).
The paper does describe prescribing practice in this patient cohort and the authors do describe the limitations of the study. However, the paper adds very little, if anything, to the current body of knowledge regarding this area.
Minor comments
When ‘H’ is first used, e.g. H+4-6, H=6-12, ‘H’ needs to be defined
What does the PT ratio mean when written as a figure <100%? It is usually written as a whole number to one decimal place e.g. 1.5. Please can the authors clarify.
The authors use the term ‘physicians’ and ‘aesthesiologists’ interchangeably which is confusing. All bar 5 respondents were anesthesiologists so it might be more appropriate to use ‘physician’ throughout and the readers will realise that this is mainly referring to anesthesiologists.
POD when used for the first time needs to be defined.
There are a number of typos and the authors need to carefully proof read the paper.
Author Response
The authors describe an online survey sent to anaesthesiologists and intensivists in France, over a 3 month period, regarding their opinion on a number of questions related to the use of LMWH in patients in the post operative period following cardiac surgery. The authors wished to obtain such opinions as the evidence regarding how anticoagulation should be managed in this period is unclear
The methodology is appropriate and well described and the study does indicate divergent practice among the small sample size that was assessed as a proportion of the number of professionals sent questionnaires (27% of the cohort).
The paper does describe prescribing practice in this patient cohort and the authors do describe the limitations of the study. However, the paper adds very little, if anything, to the current body of knowledge regarding this area.
-> We would like to thank reviewer #2 for his/her comments and insights regarding our manuscript. Following your recommendation, the manuscript has been English proofed by native English speaker. We believe this great improved both the clarity and the reading of the manuscript.
We understand the point of the reviewer regarding the novelty of this study however we would like to point out several merits of our work. First, this is the first and unique survey regarding the management of anticoagulation in post cardiac surgery patients that reports the real life usage of LMWH. We believe this is important because it shows that practice patterns notably diverge from the guidelines. There is an actual gap between practice and guidelines. This should encourage larger studies and an active dissemination of the guidelines endorsed by cardiothoracic societies. Second, the best evidence available to date remain large observational studies, and this may not be sufficient to change practice, at least this what our survey demonstrated.
Observational studies are always a good starting point because they provide a picture of the actual practice. Currently, we 're designing a study comparing different anticoagulation strategies in the immediate postcardiac surgery period.
Minor comments:
When ‘H’ is first used, e.g. H+4-6, H=6-12, ‘H’ needs to be defined -> we corrected for hours accordingly and throughout the manuscript
What does the PT ratio mean when written as a figure <100%? It is usually written as a whole number to one decimal place e.g. 1.5. Please can the authors clarify.
-> In France, PT is rather used than INR for patients without VKA medication and the prothrombin ratio is defined as the prothrombin time test plasma divided by the mean normal prothrombin time. This has been clarified in the manuscript (P6L10)
The authors use the term ‘physicians’ and ‘aesthesiologists’ interchangeably which is confusing. All bar 5 respondents were anesthesiologists so it might be more appropriate to use ‘physician’ throughout and the readers will realise that this is mainly referring to anesthesiologists.
-> We thank the reviewer for this suggestion. We corrected for physicians throughout the manuscript
POD when used for the first time needs to be defined.
-> the term has been defined on its first use
There are a number of typos and the authors need to carefully proof read the paper.
-> the paper has been proof read as requested
Reviewer 3 Report
In this manuscript, the authors summarized and discussed the Practice patterns of antithrombotic therapy in the early postoperative course of cardiac surgery. On this basis, they investigate the real-life use of LMWH in the context of early postoperative cardiac surgery among French cardiac anesthesiologists and intensivists through an online survey. This work proposes that LMWH was inconsistently used after cardiac surgery, further research is warranted to provide high-quality evidence regarding the benefits and safety of LMWH use early after cardiac surgery. Generally, the topic is interesting and the article is informative. However, the current manuscript is poorly written and must be greatly improved in its scientific presentation. I have included specific comments below that I believe may help strengthen the manuscript.
Several errors are to be found:
1、Some statements should be consistent, throughout the manuscript, (e.g., Page 1 line 27. 85% (not: Eighty five percent).
2、It is recommended to add appropriate references in the second sentence of the first paragraph of the Introduction
3、The size of words for horizontal and vertical coordinates is need to be same among figure1、2 and 3.
Author Response
In this manuscript, the authors summarized and discussed the Practice patterns of antithrombotic therapy in the early postoperative course of cardiac surgery. On this basis, they investigate the real-life use of LMWH in the context of early postoperative cardiac surgery among French cardiac anesthesiologists and intensivists through an online survey. This work proposes that LMWH was inconsistently used after cardiac surgery, further research is warranted to provide high-quality evidence regarding the benefits and safety of LMWH use early after cardiac surgery. Generally, the topic is interesting and the article is informative. However, the current manuscript is poorly written and must be greatly improved in its scientific presentation. I have included specific comments below that I believe may help strengthen the manuscript.
-> We would to thank the reviewer for his/her comments. The manuscript has been edited by a scientific native English speaker person. We believe this greatly improved the clarity and reading of the manuscript.
Several errors are to be found:
Some statements should be consistent, throughout the manuscript, (e.g., Page 1 line 27. 85% (not: Eighty five percent)
-> All similar statements about numbers and percentages have been corrected along the manuscript.
It is recommended to add appropriate references in the second sentence of the first paragraph of the Introduction
-> Several appropriate references have been added. And all the reference numbers have been updated as well.
The size of words for horizontal and vertical coordinates is need to be same among figure1, 2 and 3.
-> the X and Y-axis titles on the figures have been updated so the fonts are now the same within and between figures. We apologize for this issue.
Round 2
Reviewer 2 Report
The authors have justified in their response to this reviewer why they believe this paper should be accepted for publication - namely that it highlights that guidelines are not being followed, there is divergence in practice across France, and that these points indicate the need for additional research.
There are still some minor grammatical errors;
Page 4, line 133, change 'looked after the chest tube drainage' to 'assessed chest tube drainage'
Page 15, Conclusions, Line 53, remove 'and'
Author Response
The authors have justified in their response to this reviewer why they believe this paper should be accepted for publication - namely that it highlights that guidelines are not being followed, there is divergence in practice across France, and that these points indicate the need for additional research.
There are still some minor grammatical errors;
Page 4, line 133, change 'looked after the chest tube drainage' to 'assessed chest tube drainage'
-> The sentence has been corrected
Page 15, Conclusions, Line 53, remove 'and'
-> the word has been removed